# Impact of a Brief Family Skills Training Programme (“Strong Families”) on Parenting Skills, Child Psychosocial Functioning, and Resilience in Iran: A Multisite Controlled Trial

**DOI:** 10.3390/ijerph182111137

**Published:** 2021-10-23

**Authors:** Karin Haar, Aala El-Khani, Gelareh Mostashari, Mahdokht Hafezi, Atoosa Malek, Wadih Maalouf

**Affiliations:** 1Prevention, Treatment and Rehabilitation Section, Drug Prevention and Health Branch, Division of Operations, United Nations Office on Drugs and Crime (UNODC), Wagramer Strasse 5, A-1400 Vienna, Austria; aala.el-khani@un.org; 2United Nations Office on Drugs and Crime, Field Office I.R. of Iran, P.O. Box 15875-4557, Tehran 1994715311, Iran; gelareh.mostashari@un.org (G.M.); mahdokht.hafezi@un.org (M.H.); atoosa.malek@un.org (A.M.)

**Keywords:** parenting skills, Iran, strong families programme, resilience, mental health

## Abstract

Caregivers have a key role in protecting children’s wellbeing, and, with appropriate skills, can prevent a multitude of negative social outcomes, particularly in challenged or humanitarian settings. Accordingly, the Strong Families programme was designed as a light touch family skills programme, with a focus of supporting caregiving during stressful situations. To evaluate the short-term impact of the Strong Families programme, we performed a time-convenience, randomized, controlled trial in Iran. A total of 292 families (63% from Iranian decent, 39% from Afghan decent, and 1% other), with children aged eight to twelve years, were recruited through ten centers in Iran and allocated to an intervention (*n* = 199) or waitlist/control group (*n* = 93). The two groups did not differ demographically at baseline. We assessed families prospectively, through three scales, PAFAS (parenting and family adjustment scales), SDQ (strengths and difficulties questionnaire), and CYRM-R (child and youth resilience measure). Caregivers in the intervention group improved (highly) statistically significantly on all but one PAFAS subscales (parental consistency, coercive parenting, positive encouragement, parental adjustment, family relationships, and parental teamwork), which was not noted in the waitlist group. On the SDQ, there were (highly) significant positive changes in scores in the intervention group on all sub-scales and the “total difficulty scale“, whereas the waitlist/control group also improved on three (prosocial, conduct problems, and hyperactivity) of the five SDQ subscales. Children originating from Afghanistan improved significantly on the overall resilience scale of the CYRM-R in the intervention group, but not in the waitlist/control group. Overall, all our stratified results of the different scales reflect an accentuated improvement in families with higher levels of problems at baseline. Our comparative results indicated a strong alignment of the strong families programme with its intended short-term impact, per its logical frame on parenting practices and family management skills, children behaviour, caregivers and children mental health, and capacity to cope with stress. We postulate that the potential nudging or diffusion of knowledge (cross-contamination between intervention and waitlist/control group) at the community level could explain improvements in the waitlist/control group on some indicators, however, further research on this is recommend.

## 1. Introduction

### 1.1. Child Mental Health and Caregiver Support in Challenged Settings

Children living in humanitarian or challenged settings (such as refugee, conflict/post-conflict settings, or underserved areas) are at a greater risk of different vulnerabilities, including mental health and behavioural challenges [1,2].

The World Health Organization (WHO) defines good mental health as “a state of well-being, in which the individual realizes his or her own abilities, can cope with the normal stresses of life, can work productively and fruitfully, and is able to make a contribution to his or her community” [3,4]. Moreover, almost half of all mental disorders are initiated prior to the age of 14 years [5]. Such mental health issues often disrupt the achievement of developmental competencies and task production in youth, and, in turn, are linked to social and economic inequality, as well as increased morbidity and mortality [6]. Accordingly, the WHO has highlighted the importance of promoting good mental health in children and youth, as well as the value of developing strategies to integrate mental health promotion, as key foundations of global primary healthcare.

Many unaddressed vulnerabilities in youth are associated with poor health and developmental outcomes, such as poor mental health, violence, lower educational achievement, and substance use [7,8]. The vulnerability faced by children is further increased by family instability or poor caregiver mental health, due to prolonged periods of stress. A key factor in preventing psychological morbidity in children affected by living through challenged settings may be parental monitoring and support [9,10,11]. Primary caregivers have a critical role in protecting their children’s mental health in challenging contexts [12]. Thus, for families living in challenged settings, parental and family factors are even more significant for children to achieve positive outcomes [4,13].

While positive, nurturing caregiving can act as a protective shield, buffering negative effects on children’s well-being, conflict and displacement can compromise parental well-being and positive parenting practices, which can directly become a source of risk for children [14]. Parents experiencing high stress are less likely to provide children with the various positive interactions that promote healthy psychosocial and physical development [15]. They are, instead, more likely to engage in harsh parenting, increasing children’s risk of a variety of enduring emotional and behavioural problems [15,16]. Research with populations affected by conflict, such as in Lebanon [17] and Northern Uganda [18], suggests that caregivers’ own mental health challenges, due to exposure to war, are associated with increased child maltreatment, which, in turn, may cause an elevated risk of child mental health problems. Longitudinal studies in Afghanistan, Pakistan, and Sierra Leone have highlighted the impact of conflict exposure and how family variables, particularly the care provided by primary caregivers, continues to affect the next generation, continuing to predict the mental health outcomes in children, over and above actual traumatic experiences [19].

### 1.2. Family Skills Programmes

Interventions encouraging safe and nurturing relationships between caregivers and their children can prevent several negative social outcomes (including drug use, child maltreatment, and poor mental health) and reduce childhood aggression [20]. Family skills programmes provide caregivers with the knowledge to apply positive parenting skills. These are a set of skills that allow parents to cope and adapt to the different challenges that arise with parenting children. Opportunities for practicing these skills, through competency enhancement and support, are key to the success of such programmes [21]. These primary prevention programmes aim to strengthen the bond and attachment between caregivers and their children, by strengthening parenting skills that build key family protective factors, including communication, trust building, problem-solving skills enhancement, and conflict resolution.

The UNODC WHO International Standards on Drug Use Prevention [22], INSPIRE initiative to end violence against children [23], WHO-led Violence Prevention Alliance [20,24], and WHO/UNICEF Helping Adolescents Thrive initiative [4] have all listed and recommended evidence-based family or parenting skills programmes, as a common denominator intervention, serving multi-outcome initiatives. While there is much evidence of the effectiveness such interventions in high income and stable contexts, suggesting the potential of such programmes in other settings [25,26], the current evidence in lower-income countries is evolving and being elucidated [27].

In response, The United Nations Office on Drugs and Crime (UNODC) has been implementing a global initiative on prevention that has been piloting evidence-based family skills prevention in low- and middle-income countries globally [28]. This initiative has recently evolved in developing the Strong Families programme, as a selective, brief, family skills prevention intervention, designed to improve parenting skills, family resilience [29], as well as child well-being and family mental health. The Strong Families programme is aimed at families with children aged between eight and fifteen years living in stressful settings (including in challenged and humanitarian settings). The goal of the Strong Families programme is to support families in both recognising their strengths and skills, and further building on their strengths by sharing their challenges, as well as the things that work for them.

The Strong Families programme was drawn from three overarching theories, which shaped the components of the programme sessions and logic model. Firstly, the biopsychosocial vulnerability model [30], which suggests that positive family coping skills, such as conflict resolution, active problem-solving skills, and positive communication, shield individual family members and protect youths’ vulnerability from the negative effects of family conflicts. In this theory, caregivers’ influence on their children is greatest when the children are younger and decreases significantly as they enter early adolescence. This theory has led to a focus throughout the child, caregiver, and family sessions of the Strong Families programme, to direct attention to improving the interactions between caregivers and their children, rather than implying that either the caregiver or the child might solely hold the responsibility for any challenges or improvements. Sessions include interactive activities and role plays that provide an opportunity for families to practice positive communication, a deeper understanding of the roles and responsibilities of each family member, and appreciation for each other. The second theory is the resiliency model [31], which emphasises the foundational role that caregivers in a family play in children developing resilience. Resilience is defined as the ability to rebound from difficult or adverse circumstances [32] and is thought to develop for children more likely when raised in a family environment, in which caregivers are both positive and supportive [33]. This theory focuses on life skills that are promoted when caregivers are supportive, such as reflective skills, emotional management skills, and the ability to problem solve. This theory is supported by research that identifies that the relationship a child has with their caregiver can have a more significant impact on their mental health projectory than from the experiences of war and displacement [34]. This model identifies that resilience is fostered by the ability to manage stress; thus, the activities in the Strong Families programme focus on developing stress management strategies for all members of the family. The programme works to normalize, to caregivers and their children, the emotional and physical responses they may experience, in regards to the challenges they have been through and may still be facing. In their individual sessions, both caregivers and children are taught various stress management strategies, such as deep breathing (to regulate stress), and provided an opportunity to practice these strategies together in the family session. The third theory is social learning theory [35], which proposes that children’s daily experiences with the world, through their interactions with others and the reinforcement they receive, shape their behaviour both directly and indirectly [36]. This places the role of caregivers as pivotal for their healthy social developmental and guides family skills interventions to focus on improving the quality of parenting, by improving foundational parenting skills [37]. For example, the Strong Families programme’s caregiver sessions teach caregivers, through practical activities and role play, the skills of differential reinforcement for minor child misbehavior. This is a behaviour management strategy of rewarding the behaviour children exhibit that caregiver would like to see more of, whilst providing minimal attention to behaviour that the caregivers deem inappropriate.

The Strong Families programme was first piloted in Afghanistan [38] and has since been implemented with families in seven additional countries, in four different continents. Research findings, so far, from single-arm implementation studies in Afghanistan [38] and Serbia [39], have indicated that the Strong Families programme was feasible in low-resource and challenged settings and can be delivered by trained lay facilitators.

### 1.3. Aims and Objectives

The main aim of the current study was to expand the experience with the Strong Families programme, which had, so far, tested on its feasibility and effectiveness with one-arm research modality [38,39], through a two-arm intervention/control trial, that would further assess its impact on the different domains of its logical framework (Figure 1). The dimensions of impact to be evaluated were, namely, child mental health, parenting, family adjustment skills, and child resilience. Furthermore, such an impact would be assessed by different family characteristics at baseline, namely by including a sub-analysis of the “most-at-risk” families at baseline.

The secondary objective was to also assess the replicability of our previous single-arm pilot study in another country, Iran. Coincidentally, and while not originally intended, one further objective of this study was to compare the effect of the programme on different ethnicities (cultural backgrounds) within Iran by stratifying the analysis of results for families from Iran and those who have migrated from Afghanistan to Iran in the past. This last objective would help assess the replicability component, regardless of cultural background.

From our partner organisations, we had knowledge that families with Afghan origin had more children and that these children would also often be enrolled at a later age into school and, hence, would be older when recruiting children from the same grade into our study. To test for this hypothesis, we included an analysis to see if a higher number of children within a family, which could potentially be a more stressful situation within the home, or children with an older age would cause higher scores at baseline.

### 1.4. Country Context

Iran is a country with a population exceeding 80 million. Historically, low- and middle-income countries, such as Iran, have hosted the greatest number of refugees in the world [40]. At the time of data collection, between November 2019 and January 2020, there were an estimated 3 million documented and nondocumented Afghan refugees in Iran. The Afghan situation in Iran is typically that of prolonged exile, with large numbers residing in urban areas [41]. Reports suggest that over 80% of Afghans in Iran have access to a durable, sheltered living space with minimum standards of living [42]. Where a study of livelihood of Afghan refugees in Iran had found half of the households to have only poor income; of those, only 60% were not deprived from minimal education, health, and standards of living, based on the multidimensional poverty index (MPI) that is used for capability analyses of poverty criteria [43].

Reports indicate that in 2006, around 1.2 million documented Afghans remained in Iran, half of whom were second generation [44]. A second-generation Afghan is defined as either (1) an Iran-born individual with at least one Afghan-born parent or (2) an Afghan youth who immigrated to Iran before the age of seven. Second-generation Afghans often have different values, ideals, and beliefs, as compared to those their age from their home country or their parents. Notwithstanding, there is some similarity in second-generation attitudes and preferences, in relation to gender relations and roles, the value of education, and economic aspirations [45]. In addition, reports indicate a lower rate of involvement with drug use in Afghan children, compared to Iranian children [46].

The results from a study with 200 students in Tehran (2012–2013) indicated that positive parenting style and child-friendly styles predicted negative aggression among primary school children [47]. Further research on family functioning, conducted by utilizing a qualitative methodology with 23 Iranian parents of teenagers, showed that struggles between children and their caregivers often arose when traditional family norms do not adapt with new societal patterns [48].

Research indicates that child behavioural disorders reflect their parents’ psychological problems, as the mental illness of either parent increases the likelihood of a child presenting with a mental disorder. This is supported in a descriptive-correlational research study of 80 pre-school children registered in Tehran, Iran (2014–2015), which indicated a significant positive correlation between all dimensions of parental mental health with children’s general behavioral disorders [49].

There have been previous studies of parenting intervention implementation in Iran. For example, WHO’s international child development programme, aimed at children aged 3 years and below, was found to increase positive child–mother interactions, thus contributing to better mental health in the early years of life [50]. In a further study, the Triple P parenting programme was also implemented in Iran and showed that mothers in the intervention group reported greater improvements in both parenting style and mother–child relationship than mothers in the control group [51]. Furthermore, and including the support of the aforementioned UNODC global initiative, the SFP 10–14 (Strengthening Families Program: for parents and youth 10–14) and FAST (Families and Schools Together) programmes have been previously implemented in Iran. These programmes have shown promising result improvements in family cohesion, parent–child relationships, the skillfulness of children, and mental health [52,53].

However, despite rewarding results, most of these programmes were deemed “heavy”, meaning they requiring an infrastructure and resources that are often challenging to go to scale. Additionally, most of the aforementioned programmes have royalties, copyright costs, and impediments that might further hamper the scale-up potential. Further, they often do not particularly address or focus on the thematic of parenting under stress and neither target refugee nor displaced populations.

## 2. Materials and Methods

### 2.1. Programme Intervention

The Strong Families programme is a group intervention for children and their primary caregivers, with sessions attended over 3 weeks (one session per week). Up to two parents or main caregivers attend, with a maximum of two children under their care, aged 8 to 12 years. During week one, caregivers attend a group session, with up to 12 other caregivers, for one hour. This is the caregiver pre-session; it deals with normalizing the challenges caregivers may be facing, while also teaching effective stress management techniques. During weeks two and three, the same 12 caregivers attend the programme, accompanied by their children, for two hours. Children and caregivers from each family split into two separate rooms for the first hour and take part in group ‘child’ or ‘caregiver’ sessions. Then, during the immediate second hour, all participants group together in one room for a ‘family’ session.

The caregiver session in week two focuses on practicing using both love and limits in interacting with children and the importance of listening and communicating effectively. This is achieved through role plays, interactive activities, and group discussions. In week two, children explore what ‘stress’ means and begin to normalize feelings they may experience when stressed. They also learn stress management techniques. During the family session, caregivers and children come together and take part in activities that provide an opportunity to practice positive communication, understand the sources of challenges in each other’s lives, and practice stress relief techniques together.

In week three, caregivers practice strategies to increase their influence as a parent, such as encouraging good behaviour and discouraging misbehaviour. They also learn about using directed praise and using appropriate consequences with their children. Children explore rules and responsibilities and are guided to begin thinking about setting future goals and how they might achieve these. They are also supported in thinking about the important roles their caregivers play in their lives. In the final family session, caregivers and children take part in activities that promote exploring family values and what actions they might take to show these in their daily lives. They also spend time practicing communicating appreciation for each other.

The cultural adaptation of the Strong Families programme, in the context of Iran, was assured through seven technical sessions, from April to June 2019, with representatives from the Drug Prevention Department of the Iranian Drug Control Headquarters (DCHQ), Ministry of Education (MOH), Ministry of Health and Medical Education (MOHME), State Welfare Organisation (SWO), and the following civil society organisations: the Iranian Life Quality Advancement Institute (ILIA), Society for Recovery Support (SRS), Toloo Sobh Khorshid Institute, and UNODC. Further adding to the cultural adaptation, in the Iranian context, UNODC held an advocacy meeting with the different national counterparts responsible for drug use prevention, as well as the family skills trainers, on the added value and experience of the Strong Families programme, to facilitate endorsement at the political level. The translation of the training materials and questionnaires into Farsi was conducted by a translator, selected from the roster of the UN translators and reviewed and edited by UNODC staff.

### 2.2. Trial Design, Sampling, Eligibility Criteria, and Group Allocation

To evaluate the programme, we conducted a multisite, non-blinded, time-convenience, randomized trial with two-arms to assess effectiveness: the (1) intervention group: receiving the Strong Families programme; and (2) waitlist/control group: families only receiving the Strong Families programme after the completion of all data collection points. We prospectively collected the outcome data, assessing changes in parenting skills and family adjustment in caregivers, children’s behaviour, and children’s resilience capacities. To assess feasibility and acceptability, in the Iranian context, we additionally included an embedded process evaluation. Clinical Trial Registration: ISRCTN50189190, retrospectively registered, 20 August 2021.

Sampling utilized an opportunistic ‘universal’ approach, in which research assistants recruited families from the general population, without targeting any particular risk group. Clinical diagnoses were not assessed in participants; however, during the first session, caregivers were provided with leaflets of information on where they might access help, in case they observe severe stress reactions or any other physical, mental, or sexual health concerns they might have for themselves or their children. Inclusion criteria in the programme and the study included speaking Farsi, willing to take part in the programme, and being in the town for the duration of the whole study and measurement meetings. Families were excluded from the study if they had taken part in any other family skills training programme during the last 24 months or if the caregiver and child lived separately. Non-biased allocation to the intervention or waitlist/control group was performed only after the first data collection by convenience, i.e., availability of the families. Participants in the intervention group were then told to attend the first programme session on the same day, whereas families in the waitlist/control group were informed to attend the next data collection point 5 weeks later (2 weeks after the intervention group had completed the programme).

### 2.3. Procedure

Ten centers were nominated for the implementation of the programme in Tehran and Karaj in Iran. The ten centers were: two schools selected by the Ministry of Education (MoE), two community centers selected by the State Welfare Organisation (SWO), two community primary health care centers selected by the Ministry of Health (MoH), and four centers from non-governmental organisations (NGOs). The two schools included in the trial were the Aeme Athar elementary school for boys and Efaf elementary school for girls, both located in district 15 of Tehran, a known low-income urban area. Moreover, two centres from ILIA NGO that routinely provide educational services for children of Afghan refugees and Iranian citizens were chosen, as does SRS. The remaining centres were the Akbari Health Centre (MoH), the Hakimieh Health Centre (MoH), Ghasedak (SWO), Ghoncheh (SWO), and Toloo e Sobh e Khorshid (NGO). Overall, the centres were selected based on the main criteria of having access to families and the provision of two rooms for the programme to run and were not segregated based on gender.

Facilitators were selected based on their previous experience in school-based prevention activities, such as SFP 10–14 and FAST, or those who were familiar with prevention activities, such as SRS and ILIA, who have launched life skills programmes earlier in Iran. Facilitators were frequently staff of the centers, in order to ensure sustainability and continuation after the termination of the study. Most of the facilitators nominated by the MOH had no previous experience facilitating prevention programmes. In addition, DCHQ requested to select facilitators that have at least 10-15 years left until their retirement and separation from the institution, to ensure the retention of knowledge and skills. In November 2019, 41 facilitators (37 females and 4 males) from all 10 centers were trained on the Strong Families programme in Tehran by two international trainers. In addition, 26 research assistants (23 female and 3 male) from all 10 centers were trained by local UNODC staff (also previously trained by the same international trainers) in a 2-day training. This 2-day session covered recruitment of participants, data collection, and acquaintance with the data collection tools.

Immediately after completion of the trainings, research assistants distributed brochures containing information to the caregivers of all children aged 8 to 12 years within the reach of their respective centers. Further, they were provided with posters and banners of Strong Families to hang up on the walls of the centers to attract interested families. For the formal schools, however, two entire classes were chosen for programme implementation.

Caregivers were invited, via a self-referral process, to attend an information session, where they were given further verbal and written information and questions were answered. Although centers were instructed on the random allocation of participants through sealed envelopes, none of them complied with this. The usual practice was that after caregivers were introduced to the programme, the centers presented them with the timelines of the three cycles of the programme and caregivers could register, based on their availability. Once families agreed to take part in the study, they attended a baseline measurement session, in which written informed consent was obtained, prior to data collection. Children completed assent forms, and caregivers completed consent forms. These forms recognized that all participants had an opportunity to ask any questions about the study that they might have, did not feel any pressure to take part, and that they accepted that the data collected would be anonymized and used in scientific publications.

We enrolled families into the study between 18 November 2019 and 7 December 2019, as shown in Figure 2. We delivered the programme in all 10 centers, through two separate intervention groups, consecutively. The only rationale for this was to ensure availability of facilitators, besides their routine work, to cover all intended families. There were no differences between facilitators or allocation of families to either intervention group. The first measurement meetings took place in November/December 2019 in all study sites for intervention group 1 and in late December/January 2020 for intervention group 2 and the waitlist/control group. All families in intervention group 1 took the programme in December, and those in intervention group 2, in late December/early January, whereas families in the waitlist/control group were told to come back only for the second measurement meeting, 5 weeks later (which was 2 weeks after the completion of the intervention of intervention group 2). The initial plans were for the waitlist/control group to receive the intervention in March 2020, after all data collection was completed. However, the programme delivery in almost all centers was postponed, due to the COVID-19 lockdown measures. At the time of writing this article, Iran was still experiencing lockdown restrictions. While the commitment to make the Strong Families programme available to families on the waiting list will materialize, as soon as the measures ease, UNODC availed, in interim, a booklet and leaflet containing information, regarding parenting under COVID-19 to all families.

### 2.4. Sample Size

Based on the experiences, made with the PAFAS scores in Afghanistan, before and after the Strong Families programme delivery [38], and to show similar effects over time, a sample size of 59 was considered the minimum number of participants to be enrolled, keeping the power at 90% and the 2-sided confidence interval at 95% [55]. To compare the effect sizes in each group, a sample size of 59 families in the intervention group and 59 in the waitlist/control group seemed appropriate (See Appendix A, Table A1).

### 2.5. Data Collection

Data on the family demographics, emotional and behavioural difficulties of children, parental skills, and family adjustment measures were collected from caregivers, through self-administered questionnaires, whereas social-ecological resilience was self-reported through children. The family demographic questionnaire (FDQ) was completed at t1 (i.e., 1 week before intervention delivery), in order to collect baseline characteristics. Three outcome measures were also collected at t1, and then repeat collection of these measures was performed 2 (t2) and 6 weeks (t3) after intervention delivery. The outcome measures of concern were the paper-based parenting and family adjustment scales (PAFAS), the strengths and difficulties questionnaire (SDQ), and the child and youth resilience measure (CYRM-R). For the waitlist/control group, the same FDQ and outcome measures were taken at the same timepoints. However, no intervention was delivered in between. All families in the intervention and waitlist/control group were unaware of their group allocation when filling in the measures at t1.

#### 2.5.1. Family Demographic Questionnaire (FDQ)

The standardized FDQ had been previously used in other country contexts, such as Afghanistan [38], Uzbekistan, Zanzibar, Serbia [39], the Philippines, and the Dominican Republic, and only minor changes in some of the questions were made to reflect the Iranian context. Our aim to collect demographic questions was to perform stratified analyses, as indicated in the aims of our study. We further aimed to use the data collected through the FDQ, as a means of achieving a representative sample, by comparing baseline characteristics, such as age and gender of the caregiver, marital status, education, partner’s education, work status, partner’s work status, country of origin, years living in Iran, number of children living in the family, age, and gender of the child taking part in the programme, as well as the relationship to the child between the intervention and waitlist/control group.

#### 2.5.2. Parenting and Family Adjustment Scales (PAFAS)

The PAFAS is a 30-item questionnaire that measures parenting practices and family functioning, which are known risk or protective factors for child emotional or behavioural problems. PAFAS aims to assess changes in parenting skills and family relationships. It consists of two scales: (i) parenting, measuring parenting practices (e.g., descriptive praise, logical consequences, i.e., *“I give my child a treat, reward or fun activity for behaving well”*) and the quality or parent–child relationship (e.g., level of reciprocal warmth and parental satisfaction with the relationship to the child, i.e., *“I have a good relationship with my child”*); and (ii) family adjustment, measuring parental emotional adjustment (e.g., level of stress, depression, and anxiety experienced by a parent, i.e., *“I cope with the emotional demands of being a parent”*), as well as positive family relationships (e.g., supportive and conflict-free family environment, i.e., *“Our family members help or support each other”*) and parental teamwork (e.g., social support received from the partner in the parenting role, i.e., *“I work as a team with my partner in parenting”*). Each PAFAS question/statement can be answered on a scale from 0 (“Not true of me at all”) to 2 (“True of me very much, or most of the time”), with some of them being reverse-scored and with higher scores indicating lower levels of parenting and family adjustment skills. The PAFAS measure has shown good internal consistencies in two different Australian samples (ranging from 0.70 to 0.87), with satisfactory construct and predictive validity [56]. In addition, PAFAS has been validated in various differing cultures, such as Panama [57] and China [58], and was, more recently, utilized with Arabic-speaking families experiencing political conflict in the West bank [59].

At the time of writing this study, there were no clinically relevant cut-off points available. For sub-analyses purposes, a cut off at the 66th percentile was assumed. Participants with scores above the upper third at baseline represent those with higher levels of difficulties, and, for better readability, we call them the “most-at-risk families” throughout the text. Through the PAFAS, we intended to measure the potential changes in caregivers, related to the short-term impact, such as “Improved caregiver confidence in family management skills”, “Improved caregiving in parenting skills”, and “Increased capacity to cope with stress”, as outlined in the logical framework [38] (Figure 1).

#### 2.5.3. Strengths and Difficulties Questionnaire (SDQ)

The SDQ is a frequently used structured brief screening tool to assess children’s behavioural, emotional, and social issues over the last six months, before completion. It contains 25 questions (i.e., *“I try to be nice to other people. I care about their feelings”, “I am often unhappy, downhearted or tearful”, “I take things that are not mine from home, school or elsewhere”,* and *“I fight a lot. I can make other people do what I want”*), which can be rated from 0 (“Not True”) to 2 (“Certainly True”), with some of the questions being reverse-scored. Out of all answers, five subscales are then calculated, indicating emotional symptoms, conduct problems, hyperactivity, peer problems, and prosocial behaviors. The total difficulties score is calculated as the sum of the four subscales, excluding the prosocial behaviours. The total SDQ score ranges from 0 to 40 points, with higher scores indicating higher levels of difficulties [60]. The SDQ is widely used and now available in over 45 languages. It is commonly utilized in the family skills literature as pre- and post-intervention measures, both in short- and long-term follow ups [38,39]. The Persian version has been previously used in Iran and has shown good psychometric properties [61,62,63] in adolescents and their parents [64,65]. We used the Farsi translation that has been distributed widely [66] and the cut-points for the 4-banded categorization of the SDQ scores to classify continuous measures into “close to average”, “slightly raised/lowered”, “high/low”, and “very high/low” risk [67]. Through the SDQ, we intended to measure potential changes in children related to the short-term impact, such as “Improved child behaviour”, “Reduced aggressive and hostile behaviour”, and “Improved mental health in children”, as outlined in the logical framework (Figure 1).

#### 2.5.4. Child and Youth Resilience Measure (CYRM-R)

The CYRM-R is a self-report measure of social-ecological resilience. It has been translated into more than 20 languages [68]. The CYRM-R is regarded as a good tool for epidemiological research, particularly when used alongside instruments screening for psychosocial stress and mental health difficulties, such as the SDQ used in this study [69]. According to the logic model of the Strong Families programme [38], with the CYRM-R, we aimed to measure the short-term impacts, such as “Reduced aggressive and hostile behaviors”, “Increased capacity to cope with stress”, and “Improved mental health outcomes in children”, which would lead to the long-term impact described [38]. The CYRM-R consists of 17 items, and different versions with a 3- or 5-point Likert scale that can be used for different age groups. In our study, we used the 5-point child version (suitable for children aged 5–9 years), with questions, such as *“I talk to my family/caregiver(s) about how I feel”, “I feel supported by my friends”,* or *“People like to spend time with me”*, which can be rated from 1 (“Not at all”) to 5 (“A lot”) [70], with additional pictorial scales of glasses of water, as previously used by Panter–Brick et al. in Syrian refugee and Jordanian host-community adolescents [69,71]. The CYRM-R has previously been used in adolescents and the Middle East [69]. The translation was provided by the developers, based on the provision through Zand et al. from their use in Iranian youth [72], and has been revised by our local researchers, in order to be closer to the language actually spoken in Iran. Obtained through the CYRM-R, the overall score of resilience ranges between 17 and 85 points, deriving from the two subscales of the measures personal resilience (range 10–50 points) and caregiver resilience (range 7–35 points), with higher scores indicating a higher level of resilience. As resilience is likely to vary between contexts, no cut-offs or thresholds have been recommended by the developers [68]; however, as suggested, we separated children with low scores at t1 (≤33rd percentile) from those above in the analysis, and for better readability, we call them “most-at-risk families” throughout the text.

#### 2.5.5. Facilitator Reflection Sheets and Observer Checklist

The implementation process was evaluated based on the methods and fidelity assessment sheets previously described [73] and used for the Strong Families programme implementation in Afghanistan [38]. Coordinators in the field provided data on the number of sessions, whereas facilitators and independent observers provided information, as indicated in Appendix A, Table A8.

### 2.6. Statistical Analyses

All data were entered in Epidata version 3.1 and analysed using SPSS (version 26; IBM, Armonk, NY, USA). Plausibility checks were performed, and data completeness was assured prior to analyses. We did not impute data for the outcome variables of the three scales, as it was considered valid to ignore missing data [74]. The normality of data distribution on our multi-item Likert-type scales was assured through visual inspection of the histograms, Normal Q-Q plots, box plots and Kolmogorov-Smirnov tests. Cronbach’s alpha coefficients were calculated to assess internal consistency of the SDQ, PAFAS, and CYRM-R subscales at all measurement-points. Continuous variables are presented as mean and standard deviation (SD) with a 2-sample *t*-test for comparison, whereas categorical data were summarized as frequencies and proportions and compared using a chi-square test.

To compare scores at the different time points, we first tested a potential group-interaction effect through a two-way mixed ANOVA, with, within, and between subjects’ factors. We further tested the effects of the respective outcome variable for families in the intervention and waitlist/control group separately, through a repeated measures ANOVA, accounting also for potential non-homogeneity of covariances, with post-hoc tests using Bonferroni corrections. Further, for ANOVA, we made sure to test for the required additional key assumptions, such as independent variables, outliers and normality, homogeneity of variances, covariances, and sphericity. In case Mauchly’s test of sphericity indicated that the assumption of sphericity had been violated, a Huynh–Feldt correction was used. Homogeneity of Variances was tested through a Levene’s test [75]. Results are reported separately for families originating from Iran and Afghanistan. Participants with worse scores at baseline (and for better readability, called “most-at-risk families” throughout the text) were analysed separately for each of the subscales.

A multiple regression was included to determine how much of the variation in the dependent variable was explained by the independent variables. Before running the multiple regression, we checked for the key assumptions being met: independence of errors (residuals) through the Durbin–Watson statistic, linear relationship through scatterplots and partial regression plots, homoscedasticity of residuals through plotting the studentized residuals against the unstandardized predicted values, no multicollinearity through correlation coefficients and Tolerance/VIF values, no significant outliers, leverage and influential points and for errors (residuals) being approximately normally distributed through a normal Q-Q Plot of the studentized residuals. All data were analysed following the intention-to-treat approach. Statistical significance level was set at *p*-value lower than 0.05.

Clinical Trial Registration: ISRCTN50189190, retrospectively registered, 20 August 2021.

## 3. Results

### 3.1. Exclusion of Analyses, Missing Data, and Loss to Follow-Up

Two centers, accounting for 85 participants (56 in the intervention group and 29 in the waitlist/control group; Figure 2), had to be excluded from the analyses, due to low fidelity and inconsistency in data collection. According to both data sources of fidelity indicators (facilitators and independent observers; Appendix A, Table A8), most of the topics within each session were not covered in these two centers, attending caregivers changed, many sessions had less than two facilitators present and there were not one or more of the same facilitators present throughout the sessions. In addition, data were collected in an inconsistent way, with many duplications of IDs and wrong allocation to group; hence, the questionnaires could not be retrospectively assigned with certainty to the correct participants/IDs. After comparison, we regarded them as missing at random and excluded them from the analyses.

Hence, overall, 292 participants were included in the analysis, 199 in the intervention group and 93 in the waitlist/control group (Figure 2). At t2, 272 participants completed data collection (93%) and at t3, 207 participants (71%), respectively. Overall, there were slightly more missing data at t2 in the intervention group, compared to the waitlist/control group (9% versus 2.2%; χ^2^ = 4.72; 1 df; *p* = 0.03), but this did not differ by country of origin. Families who did not fill in a questionnaire at t3 were more likely to be from Iran (41.6% missing questionnaires) than from Afghanistan (7.9% missing questionnaires; χ^2^ = 36.14; 2 df; *p* < 0.001). Out of the 20 people overall who did not attend at t2, three came back again at t3.

Overall, we did not see a systematic error in families with missing follow up questionnaires. Families with missing questionnaires, compared to other families, did not differ at baseline, in terms of their country of origin or the level of problem, as registered on the outcome scales. We, accordingly, concluded that questionnaires at t2 and t3 were missing at random, as shown in Appendix A, Table A2.

Out of all 292 caregivers, there were 3 missing FDQs. Among the remaining, completion rate was good, with only few missing answers: age of caregiver: 2 missing cases, education (*n* = 2), partner’s education (*n* = 10), work status (*n* = 12), partner’s work status (*n* = 17), number of children to care for (*n* = 10), age of the child taking part in the programme (*n* = 1), gender (*n* = 1), relationship to the child (*n* = 2), place of birth of the child (*n* = 2), and country of origin (*n* = 5).

Within all PAFAS questionnaires that were filled in, some questions were left out at t1 (PAFAS 11, 13, 14, and 20, respectively, with 5.6% missing answers each), this improved, however, at t2 and t3, with >95% of questions having been completed at t2 and t3. Within the SDQ and CYRM-R, more than 95% of all questions were answered at all 3 timepoints.

### 3.2. Description of Demographics

Overall, the 292 families who completed the data collection and were included in the analysis, were recruited from eight different sites.

Families in the intervention and waitlist/control group did not differ with respect to demographic baseline characteristics, such as age and gender of the caregiver, relationship to the child, marital status, education, partner’s education, work status, partner’s work status, country of origin (63% Iran, 36% Afghanistan, and 1% Pakistan overall), years living in Iran (15.1 +/− 13.86 years), number of children living in the family, and age of child taking part in the programme, as shown in Table 1.

There were slightly less boys in the intervention group (40%), compared to the waitlist/control group (52%; χ^2^(1) = 3.898, *p* = 0.048; Table 1). Caregivers with Iranian background had a higher level of education than caregivers originating from Afghanistan (27% completing university or post-graduate degrees and 9% with primary school education or less in Iranian caregivers vs. 1% and 72% in Afghan caregivers, respectively; χ^2^(10) = 130.573, *p <* 0.001). Caregivers originating from Afghanistan had more children (2.92 +/− 1.249) than those from Iran (1.71 +/− 0.778; t_267_ = −9.745, *p* < 0.001) and Afghan children were significantly older than Iranian children (10.41 +/− 1.767 years vs. 9.38 +/− 1.507 years, respectively; t_276_ = −5.130, *p* < 0.001).

### 3.3. Parenting and Family Adjustment Skills

Both the PAFAS Parenting, and Family adjustment scales had acceptable to good levels of internal consistency, as determined by Cronbach’s alpha scores of 0.76 and 0.84 at t1, 0.74 and 0.86 at t2 and 0.74 and 0.86 at t3.

#### 3.3.1. Overall Parenting and Family Adjustment Skills Results Comparing Intervention and Waitlist/Control Group

Over time, we found (highly) significant improvements in caregivers in the intervention group on all subscales apart from “parent-child relationship”, that were not found in the waitlist/control group, as shown in Figure 3.

#### 3.3.2. Parenting and Family Adjustment Skills Results by Country of Origin

In Iranian caregivers, there was a statistically significant interaction between our intervention and time on coercive parenting scores (partial η^2^ = 0.050). This reflects that mean coercive parenting scores changed significantly differently in intervention versus waitlist/control group over time. That is, mean coercive parenting scores changed differently over time, depending on whether Iranian caregivers took the Strong Families programme or not. Likewise, in Afghan caregivers, we found a statistically significant interaction between the Strong Families programme and time on parent–child relationship scores (partial η^2^ = 0.075), as shown in Appendix A, Table A3.

When repeated measures ANOVAs was estimated separately for Iranian and Afghan caregivers by intervention groups, we found that both Iranian and Afghan caregivers receiving the intervention improved over time on the dimension of “coercive parenting” (partial η^2^ = 0.196 and 0.125) and the family adjustment scales “parental adjustment” (partial η^2^ = 0.104 and 0.074), “family relationships” (partial η^2^ = 0.180 and 0.112) and “parental teamwork” (partial η^2^ = 0.101 and 0.061). Such a statistically significant change was not noted in the waitlist/control group over time. Detailed statistical results can be found in Appendix A, Table A3.

#### 3.3.3. Parenting and Family Adjustment Skills Results in Caregivers above the 66th Percentile (“Most-at-Risk” Families)

In the two-way mixed ANOVA, we found a statistically significant interaction between our intervention and time on the “coercive parenting” (partial η^2^ = 0.057), “parent-child relationship” (partial η^2^ = 0.063), and “family relationships” (partial η^2^ = 0.052) subscales in caregivers with scores above the 66th percentile at baseline (*“most-at-risk” families*). Scores on these three subscales changed differently over time, depending on caregivers having taken the Strong Families programme or not. When testing with repeated measures ANOVA independently for each group, in all three dimensions mentioned above, caregivers in the intervention group improved significantly over time (with partial η^2^ ranging from 0.334 to 0.412), which was not found in the waitlist/control group on the “coercive parenting” subscale. On the “parent-child relationship”, and “family relationships” subscales, however, caregivers in the waitlist/control group also had significant declines in scores over time (with partial η^2^ ranging from 0.313 to 0.480).

Mean scores of all caregivers with scores above the 66th percentile at baseline (*“most-at-risk” families*) showed highly significant reductions in scores in all PAFAS subscales in the intervention group (with partial η^2^ ranging from 0.167 to 0.412). However, there were also improvements in those caregivers in the waitlist/control group on the “parental consistency” (partial η^2^ = 0.334), “positive encouragement” (partial η^2^ = 0.296), “parent-child relationship” (partial η^2^ = 0.480), “parental adjustment” (partial η^2^ = 0.200), and “family relationships” (partial η^2^ = 0.313) subscales, as shown in Appendix A, Table A4.

Apart from “family relationships”, where caregivers’ scores in the intervention group decreased significantly over time, there was no statistically significant effect on scores in neither the intervention nor waitlist/control group caregivers that had scores below the 66th percentile at baseline (data not shown).

### 3.4. Child Mental Health

The reliability of the total difficulties score indexed by Cronbach’s alpha was regarded acceptable to good at all 3 timepoints (0.77, 0.77, and 0.80).

#### 3.4.1. Overall SDQ Results Comparing Intervention and Waitlist/Control Group

Overall, there were (highly) significant changes in scores in the intervention group on all subscales and accordingly on the “total difficulty scale”. In the waitlist/control group the overall SDQ score also significantly improved over time and we noted the improvement on three of the five SDQ subscales (prosocial, conduct problems and hyperactivity). However, no significant change over time was noted on the “emotional problem scale“ or the “peer problem scale“ in the waitlist/control group, as shown in Figure 4.

#### 3.4.2. Child Mental Health Results by Country of Origin

Analyzing children of Afghan origin and Iranian origin separately, there was no statistically significant interaction between the intervention and waitlist/control group and time on any of the SDQ subscales or the total difficulty scale.

In general, children from families originating from Afghanistan started off with higher scores in both, the intervention and waitlist/control group. While reduced over time, the mean scores at t3 in both groups remained higher in Afghan children, compared to those from Iranian families.

Overall, apart from the hyperactivity scale, there was no effect in Iranian children in the waitlist/control group over time, in any of the subscales, nor the total difficulty score. However, Iranian children in the intervention group improved significantly on the emotional problem (partial η^2^ = 0.112), conduct problem (partial η^2^ = 0.062), peer problem (partial η^2^ = 0.065), and total difficulty scale (partial η^2^ = 0.218), as assessed through repeated measures ANOVAs for each subgroup. Scores in Afghan children improved significantly over time in both intervention groups on the conduct problem, hyperactivity, prosocial, and total difficulty scales, whereas on the peer problem scale, only children in the intervention group improved, but not those in the waitlist/control group, as shown in Appendix A, Table A5.

A multiple regression was run to determine how much of the variation of the “total difficulty score” was explained by the number of children in the family, the age of the children and the country of origin. The multiple regression model statistically significantly predicted the “total difficulty score”, F(3,221) = 9.990, *p* < 0.001, adj. R^2^ = 0.107. However, only the country of origin added statistically significantly to the prediction, *p* < 0.001. Regression coefficients and standard errors can be found in Appendix A, Table A6.

#### 3.4.3. Child Mental Health Results in Children with High and Very High Total Scores at Baseline (“Most-at-Risk” Families)

We did not find an interaction between the intervention and time on the “total difficulty scale” in children with high and very high scores (17+ points; *“most-at-risk” families*). However, when both groups were analyzed separately, there was a significant effect over time, decreasing from 20.7 (+/− 3.31) at t1 to 16.9 (+/− 4.90) at t3 consistently in the intervention group (F(2,68) = 16.05, *p* < 0.001, partial η^2^ = 0.321) and from 20.0 (+/− 2.90) to 15.0 (+/− 4.59) in the waitlist/control group (F(2,30) = 14.31, *p* = 0.003, partial η^2^ = 0.329). In children below 17 points at baseline, we saw a significant decrease in the intervention group overall (F(2,122) = 9.404, *p* < 0.001, partial η^2^ = 0.134), whereas there was no effect in the waitlist/control group.

#### 3.4.4. Child Mental Health Results by Gender

Comparing boys only, we found a significant improvement over time on the “total difficulty scale” in those in the intervention group (F(2,86) = 9.83, *p* < 0.001, partial η^2^ = 0.186), but not in the waitlist/control group, which could similarly also be found in the “conduct problem scale”, “peer problem scale”, and “prosocial scale” (data not shown). In girls, however, both groups improved over time on the “total difficulty scale”, the intervention (F(2,104) = 11.37, *p* < 0.001, partial η^2^ = 0.179) and waitlist/controll group (F(2,46) = 6.51, *p* = 0.003, partial η^2^ = 0.220). In girls, both groups improved on the “conduct problem scale”, the “hyperactivity scale”, and the “prosocial scale” over time; however, in the “peer problem scale”, only girls in the intervention group improved significantly, but not those in the waitlist/control group (data not shown).

### 3.5. Child Resilience

The internal consistency of the overall resilience as measured through the CYRM-R questionnaire reached good reliability with Cronbach α scores of 0.82, 0.81 and 0.87 at the three time points.

#### 3.5.1. Overall Child Resilience Results Comparing Intervention and Waitlist/Control Group

There was no significant difference in the intervention or waitlist/control group over time overall, as shown in Table 2.

#### 3.5.2. Child Resilience Results by Country of Origin

All children originating from Iran were already in the “high resilience” category at baseline when cautiously applying thresholds according to Canadian data as reported by the developers of the CYRM-R tool [56], whereas children from Afghan families started off in the “moderate resilience” category thereof.

We did not find an interaction between the intervention and time on any of the resilience scales in children stratified by the country of origin; however, in the repeated measures ANOVA, children of Afghan origin improved significantly on the “caregiver resilience subscale” (partial η^2^ = 0.106) and the “overall resilience scale” (partial η^2^ = 0.080) in the intervention group but not in the waitlist/control group, as shown in Table 2.

For the total SDQ, a multiple regression model significantly predicted the “overall resilience scale”; however, only the country of origin added statistically significantly to the prediction (not the number or age of children) (data not shown).

#### 3.5.3. Resilience Results in Children below the 33rd Percentile (“Most-at-Risk” Families)

In children who had scores below the 33rd percentile at baseline (*“most-at-risk” families*), there was a significant improvement over time, as found in the repeated measures ANOVA, on the “overall resilience scale” (partial η^2^ = 0.241), as well as the “personal resilience subscale” (partial η^2^ = 0.153), in the intervention group but not in the waitlist/control group. In children with scores below the 33rd percentile at baseline (*“most-at-risk” families*) on the “caregiver resilience subscale”, both, the intervention group, and the waitlist/control group improved significantly over time, as shown in Appendix A, Table A7. In all 3 dimensions, there was no effect in either group over time in children who were above the 33rd percentile at baseline, and neither in girls nor boys separately (Data not shown).

### 3.6. Process Evaluation

Positive results were seen on all process evaluation components, in addition to the respective performance indicators. On assessing the fidelity of the eight centres that were included in the analyses, 95–100% of the programme activities were covered. Very high consistency of facilitators attendance, as well as group size, was also found. Facilitators reported 100% of doses received throughout all sessions, measured through both the interest of caregivers, as well as their children. Assessing the reach of the programme, overall, 87% of families attended all three sessions of the programme. Inputs, such as the quality of childcare, travel arrangements, refreshments, rooms, materials, and equipment were rated positively, between 94% and 100% overall, as shown separately for each session in Appendix A, Table A8.

## 4. Discussion

### 4.1. Overall Effect of the Strong Families Programme

This study further expands the knowledge cumulated from previous single arm pilots of the Strong Families programme pilot in Afghanistan and with Afghan refugees in Serbia [38,39]. The previous studies indicated positive feasibility and potential effectiveness of the programme on mental health, parenting and family adjustment indicators on families living in stressful situations. This study results’ echo the previously registered initial findings, through a two-arm time-convenience randomized trial. The results focused on the short-term impact of the programme, as per its designed logical framework. The positive changes, despite the light design of the intervention, which was found feasible to implement in other resource-limited settings and through recently trained facilitators, were noted at the level of improved parenting skills, youth mental health and adolescent resilience. Moreover, while the results are seemingly positively affecting all children, it was evident that children with poorer scores at baseline benefited the most. We, therefore, conclude that improving interactions between caregivers and their children, practicing positive communication, developing stress management strategies, and learning behaviour management strategies, as conveyed through the Strong Families programme, leads to the intended short-term results, and potentially promotes healthy psychosocial and physical development of children.

### 4.2. Effect of the Strong Families Programme on Parental Adjustment and Functioning

Caregivers receiving the Strong Families programme indicated a significant improvement on three of the four parenting subscales and on all the family adjustment subscales. While the PAFAS instrument does not carry a specific threshold value to separate subgroups of parenting skills, our interest was to check effectiveness on *“most-at-risk” families.* Focusing on these families, it was encouraging to see the significant improvement favouring the intervention group, which was further supported by higher effect sizes than in the overall changes in scores. While this is a welcomed finding, it would be worth exploring in future studies if caregivers with higher problems at baseline (*“most-at-risk” families*) might be more receptive and amenable to parenting advice than those who do not feel such a need per baseline assessment.

It was also evident that certain subscale-scores were already in the upper range of the subscale at baseline (example Parental consistency, Coercive parenting, and Parental adjustment subscales), whereas others were in the lower quartile of the range (namely the parent–child relationship). Subscales that started off at higher mean scores improved the most, which is a further encouraging sign. However, in-depth research is encouraged to further assess the cultural significance or generalizability of such findings to practically guide the focus of family skills interventions on the domains that need the most attention to improve efficiency of implementation. It was interesting to note that the PAFAS mean scores at baseline of Afghan caregivers living in Iran were quite comparable to those generated from a previous study of caregivers living in three different cities in Afghanistan [38]. The subscales of Afghan ethnicities between the two countries were as follow (Parental consistency: 7.77 vs. 7.77; Coercive parenting: 8.07 vs. 8.49; Positive encouragement: 3.04 vs. 1.83; parent–child relationship: 3.57 vs. 2.03; Parental adjustment: 7.71 vs. 6.52; Family relationship: 4.56 vs. 4.31; Parental teamwork: 3.46 vs. 2.86 in Iran and in Afghanistan, respectively). This might be a potential initial insight that certain cultural or social backgrounds might be associated with particular parental faculties that could need a differential level of attention.

Nevertheless, despite the cultural and demographic differences between Afghan and Iranian caregivers participating in the study, the intervention positively affected the faculty of “coercive parenting” subscale, as well as all three family adjustment subscales. Such findings of positive parenting impact across subgroups of different cultures within the same country, carries important public health significance in terms of its application in a universal format, its potential for scale up, and for the affinity for diverse families to attend to and benefit from its sessions.

What was also worth noting was the improvement in scores in the waitlist/control group over time. The comparison group improved on the subscale of parent–child relationship; moreover, when focusing on the families scoring above the 66th percentile (*“most-at-risk” families*) across subscales, the comparison group also improved over time. It would be worth exploring in future studies if this subscale change in the comparison group might be the result of nudging the families through the invitation to participate in the research. This parallels a finding from a previous study on a light touch family skills intervention we undertook in the West Bank. In this study an improvement in parenting was found in the control group, as well as a reduction in child emotional and behavioural difficulties, potentially attributed to nudging families by invitation. This illustrates the potential value of messages on parenting in context where caregiver support is not common [59]. Accordingly engaging families in conversations on parenting might already bring about positive changes, this speaks in favor of the value of light interventions, particularly in situations of high need. Alternatively, this might be the result of waitlist/control -group contamination effect. According to the communication with Iranian NGO partners who provide services for Afghan caregivers, it is common that mothers within the same community attend common classes or activities (handicraft/beauty). During such encounters and given the community ties existing they would have the opportunity to interact with each other, including potentially in the area of exchanging knowledge of programmes or lessons they have been exposed to. Hence, we cannot exclude that caregivers, who were allocated to the intervention group shared programme content with fellow caregivers allocated to the waitlist group before completing all data collections. Unfortunately, we cannot verify this through the data collected in this study; however, we recommend more qualitative research to investigate the potential “cross-fertilisation” effect. While potentially contaminating the results on effectiveness, this remains a positive side effect of the intended objective to carry a multiplying or echoing effect of our programme through these informal networks. Further research is required to better explore such a mechanism, but it carries important public health implications, particularly for dissemination of such knowledge in migrant, displaced or refugee families where accessibility to such intervention is often complicated.

Through our comparative trial, we conclude that the intended short-term impact of the Strong Families programme regarding “Improved caregiver confidence in family management skills”, and “Improved caregiving in parenting skills” was reached, as outlined in the logic model [38]. (Figure 1) Based on the literature, these indicators also support the path to the programme desired long-term impact, which include reduced substance use, violence and risky behaviours in addition to improved mental health in caregivers and children [13,76]. Long-term follow up of a cohort would be recommended to further corroborate the long-term impact, and the planning of such is encouraged.

### 4.3. Effect of the Strong Families Programme on Mental Health

When it comes to the SDQ scale, it was also worth noting the very encouraging finding of improvement of the scores across every subscale of the SDQ in the intervention group.

It is also worth mentioning that the total SDQ score of Afghan children at baseline in this study was comparable to that of the Strong Families programme pilot implemented in 3 cities in Afghanistan (Kabul, Herat, and Balkh) back in 2018 (mean score 16 vs. 18, respectively). While the score in both instances was in the high range, there was not one single subscale that accounted for the slight difference, but small amounts on each of the subscales, resulting in the slightly higher scores in families living in Afghanistan in 2018, compared to Afghan families living in Iran in 2020. This total mean score was higher than that recorded for children of Iranian origin (total mean SDQ score of 12 at baseline). Further, cross-cultural issues could have played a role when comparing the psychometric properties of rating scales [62], accounting for the different scores between Iranian and Afghan children in Iran.

Amongst Iranian children rated on SDQ, a significant improvement was noted over time on emotional problems, conduct problems, peer problems and total difficulty SDQ score in the intervention group but not in the comparison group. What was notable was the improvement of the SDQ score amongst Afghan children on 4 of the 5 subscales in the intervention group and 3 of the 5 subscales in the comparison group. Moreover, in children with high and very high SDQ scores (17+ points; *“most-at-risk” families*) on the total difficulty scale at baseline, we saw improvements over time in both, the intervention and waitlist/control group, whereas in children below 17 points, only the intervention group improved. Much like the discussion on PAFAS results, we cannot exclude the potential “nudging” or “contamination”, “cross-fertilisation” of the waitlist/control group by the intervention group. Particularly, children that were recruited through schools were de facto randomly allocated to the intervention and waitlist/control group; however, we cannot exclude that children were talking to each other in class about the things they learnt in the programme, and children in the waitlist/control group who did not attend any programme might have been interested to know from their peers what it was all about and what they had learned. It is important to note the fact that boys in the intervention group (but not in the waitlist/control) improved on the SDQ scores on three subscales and the total SDQ scores. The cross-contamination was more prominent in girls, who improved in three of the five subscales in the waitlist/control group and four of the five subscales in the intervention. This gender difference in diffusion of knowledge, would be an important factor to assess in future studies. What was nevertheless clear was children of families receiving the Strong Families programme improved on more subscales of the SDQ that those who did not. So, while the potential cross-contamination or nudging might be occurring (a positive side-effect of the intervention), direct exposure to the programme remains seemingly more rewarding in terms of effect.

The Validity of Farsi Version of Strengths and Difficulties Questionnaire (SDQ) has been assessed in Iran [77] and is a frequently used screening tool with total difficulty scores above 17 points indicating abnormality as described previously [60] and adjusted to 19 points in an Iranian sample [62]. Applying these definitions, none of our mean scores reached any clinically apparent level; however, there were a few children with scores above 19 points at start point. The Strong Families programme was not developed to treat individual mental health disorders; however, all facilitators were trained to refer any person with additional need for support to the respective local institutions, and they also handed out leaflets with addresses of such institutions at the first encounter with families.

### 4.4. Effect of the Strong Families Programme on Child Resilience

Resilience, on the other hand, includes a wider area, such as family, friends and community that influence an individual’s capacity to cope successfully, despite substantial life challenges. Research has shown one’s successful resilience towards everyday stressors has the same components as that of post-disaster [78,79]. In our study, children originating from Iran scored around 42 points on the personal resilience and 32 on the caregiver resilience subscale at baseline. Compared to children from 14 other countries with similar age and gender-distribution, the Iranian children in our sample, were already seemingly on the better margin of the scale, compared to 40 and 28 points on average in the summarised sample [80]. Therefore, we do not find it surprising that no significant improvement was found on intervention completion, although the scores increased slightly.

Children from Afghan families, however, started off with 40 and 29 points on the respective subscales, and improved overall, particularly on the “caregiver resilience subscale” after the intervention, which stands for improved characteristics associated with the important relationships shared with either a primary caregiver or family [68]. Therefore, it can be concluded that the Strong Families programme was effective, particularly in those with lower scores at start point (*“most-at-risk” families*). This finding is further supported by the significant improvement on the “personal resilience subscale” (standing for intrapersonal and interpersonal items that are linked as both dimensions, depending on individuals’ social ecologies to reinforce their resilience [68]) and the overall resilience scale in the intervention but not the waitlist/control group in children with poor scores at baseline (*“most-at-risk” families*).

All children enrolled in this study were from disadvantaged areas (per the personal communication with the implementing bodies, NGOs, and others in the field). However, it seems that children from Afghan families might have benefitted more, for different reasons. Afghan children were on average one year older than Iranian children and were more likely to go to informal schools or were enrolled in classes one year below their fellow Iranian counterparts. As Strong Families is a family skills programme, and not a separate child or parenting programme, we believe that this might have particularly benefitted the Afghan families, who might not have been thinking proactively about their roles and responsibilities within their family context. Our facilitators and research assistants reported that some Afghan children had difficulties in understanding some of the questions used in the CYRM-R and that they assisted by reading out and providing explanations for such children (for example of the word “opportunity”). We do not think that language was a barrier in delivering the family skills programme itself, as facilitators who implemented in the respective areas were also of Afghan origin; in addition, Dari is similar to the Farsi spoken in the area.

We did not gather data on the experience of trauma in children or caregivers and hence cannot link the different mean scores at baseline to potential traumatic experiences in the past. However, as described by Panter–Brick et al., resilience is not necessarily the opposite of risk, as for example Syrian refugees experienced high trauma exposure, even when this did not equate to lower resilience scores, and vice versa for Jordanian host residents [69]. Particularly in the Middle East, families seem to be fundamental resources and enablers or barriers in matters of school, education, or employment [69]. Höltge et al., compared the same resources across different countries, described common characteristics of adolescent resilience networks. Caregiver support was identified as the most significant factor for adolescent resilience, with the strongest positive associations with other resources, across the samples 18,914 adolescents from 14 countries studied [80]. Furthermore, children exposed to (repeated) community violence have benefited from social support from a child’s family (parent), school, and peer group and factors, such as family cohesion and positive coping mechanisms of the caregivers lessened the negative impact of community violence [81,82,83,84,85]. The importance of interdependent coping has been highlighted by studies of populations living in war zones that have indicated that the most important factor in predicting a child’s response was the level of emotional upset and anxiety displayed by parents, not the war itself [81,86]. Likewise, if parents used supportive, reliable, and no punitive parenting methods, children’s aggression and depression did not increase through intensive exposure to war trauma [81,87,88].

Overall, we found the CYRM-R a user-friendly tool that was easy to understand by the children enrolled in our programme. Its previous versions had already been tested in the Iranian context, and the 11-item version of this questionnaire showed satisfactory validity and reliability in 2012/13 and was found as an appropriate tool for measuring resilience in Iranian adolescents [72].

Again, with an SDQ and CYRM-R lens, through our comparative trial, we conclude that the intended short-term impact of the Strong Families programme regarding “improved child behaviour”, “reduced aggressive and hostile behavior”, “increase capacity to deal with stress”, and “improved mental health in children”, as outlined in the logic model, was reached [38]. (Figure 1) Still, long-term follow up of a cohort would be recommended to further corroborate the long-term impact, and the planning of such is encouraged.

### 4.5. Limitations of the Study

Despite all efforts, our study also has some limitations. First, blinding of participants was not feasible, as families noticed if they underwent the training programme right after the first data collection or if they were only invited for the next appointment on data collection. However, none of the participants were aware of their allocation at the first data collection, as they were only informed after all questionnaires had been filled in. We further aimed to reduce selection and allocation bias, by allocating participants to either group only after the first data collection and purely based on their availability and their own choice of dates. Comparing the baseline demographics, we did not find any differences between intervention and waitlist/control group, and, therefore, conclude that these potential biases were minimized.

Another limitation of our trial was the impact of various serious events in the country, such as the assassination of Gen. Qasem Soleimani, the shooting of Ukraine flight 752, and air pollution, which resulted in the closure of schools and impacted our data collection, particularly at t2. Furthermore, our third data collection could not be fully completed, due to COVID-19 lockdown measures that came into effect in February 2020 in Iran. We do believe, however, that those lost to follow-up are missing at random, as the lockdown was universal in the country and families not filling data on this wave did not differ from any other family but were hampered in attending the data collection meetings, due to a force majeure.

Although our programme and invitation for participation was universally targeted, our sample was predominantly mothers. The lack of participation of fathers in parenting research and implementation is a common limitation in the parenting literature. This is even more evident in settings of humanitarian or underserved contexts. We, therefore, cannot generalise the findings to the male caregiver population. However, future research should shed light onto the role of fathers/male caregivers in family skills programmes and investigate the potential of including both caregivers in such programmes.

Finally, language skills might have played a role in families originating from Afghanistan. According to our demographic data, families had already been living on average for 15 years in Iran; however, we did not collect any data on how well-integrated they were within their local family networks or if they spoke sufficient Farsi. Based on communication with facilitators, this was not a problem in the programme delivery, as facilitators delivering the programme to Afghan families also spoke Dari and delivered the programme in Dari. All questionnaires, however, were in Farsi, and research assistants reported that they had to assist children, particularly in filling in the CYRM questionnaires. In addition, we had modified the CYRM questionnaire used by Zand et al. [72]. However, we did not back-translate it; thus, the language used might not have been clear enough, particularly for non-native Farsi speakers. When interpreting the data, we could only use the Canadian thresholds provided [68] and, hence, encourage researchers to share their international experiences in order to create global or context-specific thresholds in the future.

### 4.6. Implications for Utilizing Family Skills Programmes within the Current Political Context

In the past, it was already found difficult to reach caregivers, in humanitarian contexts, with evidence-based family skills programmes [89,90,91], despite their significant needs. As part of our secondary objective, we compared the effect of the Strong Families programme on different ethnicities, namely in families from Iran and those who have migrated from Afghanistan to Iran in the past. We have reported previously on families living in Afghanistan [38], and also on Afghan families who migrated to Europe and were stranded in reception centres in Serbia, respectively [39]. Given the current political situation, and the expected migration of many more Afghans to neighbouring countries, we strongly believe that the replicability and value of our programme should be emphasized. We found it feasible to implement the Strong Families programme through trained lay facilitators and recommend scaling up in the respective areas, should the need arise.

## 5. Conclusions and Future Directions

In its short-term follow up, and per its designed logical framework, the Strong Families programme carries notable effect on improved caregivers’ skills and confidence in family management. It increased the capacity of the family members to cope with stress, improved mental health outcomes amongst family members, and progressed children’s positive behaviors and attenuation of negative behaviors. These findings are particularly valuable, given the programme is designed to be a light-touch, low resource package making the process of its scalability, especially in resource-limited communities, easier to materialize. The findings associated to potential a “cross-fertilization” or “nudging” effect, while requiring further in-depth evaluation, stands in further support of the scalability of effectiveness and potential return on investment under limited, community-based prevention budgets. The biopsychosocial vulnerability, resiliency, and social learning theory models that guide its content indicate that such short-term changes, at the family level, are good indicators, in support of longer-term impacts on child development that requires further research, particularly on the elements of reduction of violence, substance use, and mental health outcomes (of rising concern especially during, and as an aftermath, of the COVID19 pandemic). Nevertheless, the current findings, including replicability of findings from other countries, already suggest a strong advocacy message for stakeholders implicated in child mental health, resilience, and healthy and safe development, especially those living in particularly underprivileged circumstances, to consider the Strong Families programme as a package of support.

## Figures and Tables

**Figure 1 ijerph-18-11137-f001:**
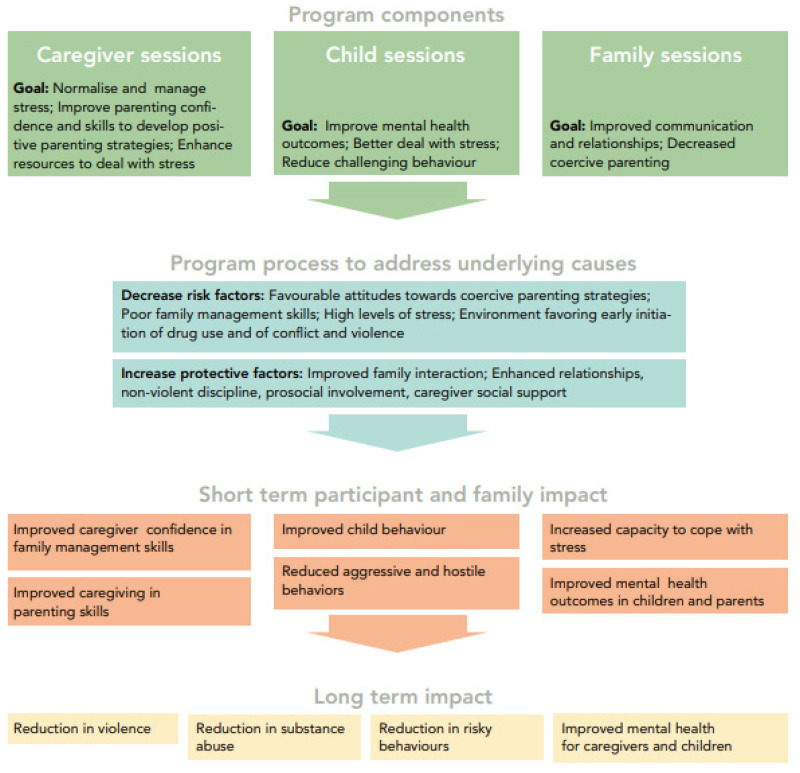
Logic model of the Strong Families programme [38].

**Figure 2 ijerph-18-11137-f002:**
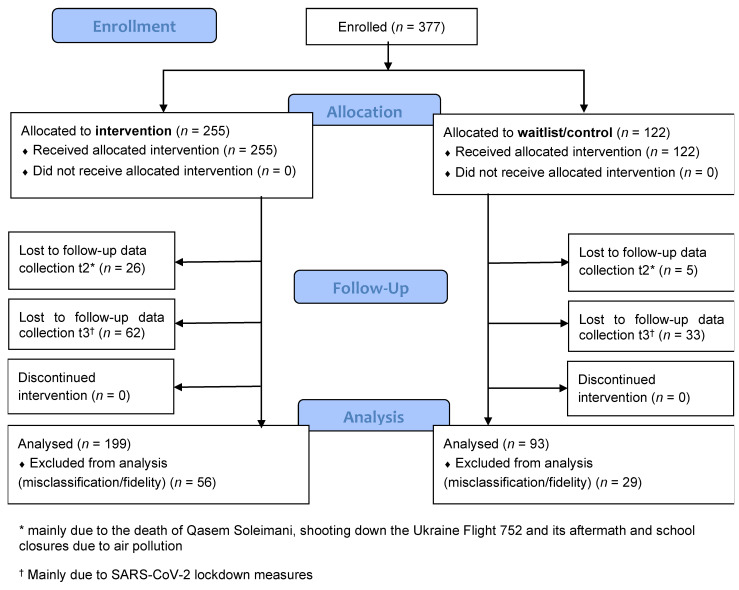
Modified CONSORT flow diagram [54].

**Figure 3 ijerph-18-11137-f003:**
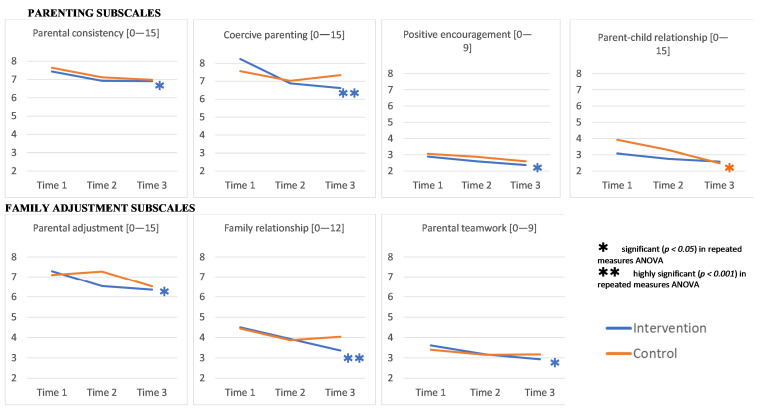
Overall parenting and family adjustment results comparing intervention and waitlist/control group (*higher scores indicating lower levels of Parenting and Family Adjustment skills*).

**Figure 4 ijerph-18-11137-f004:**
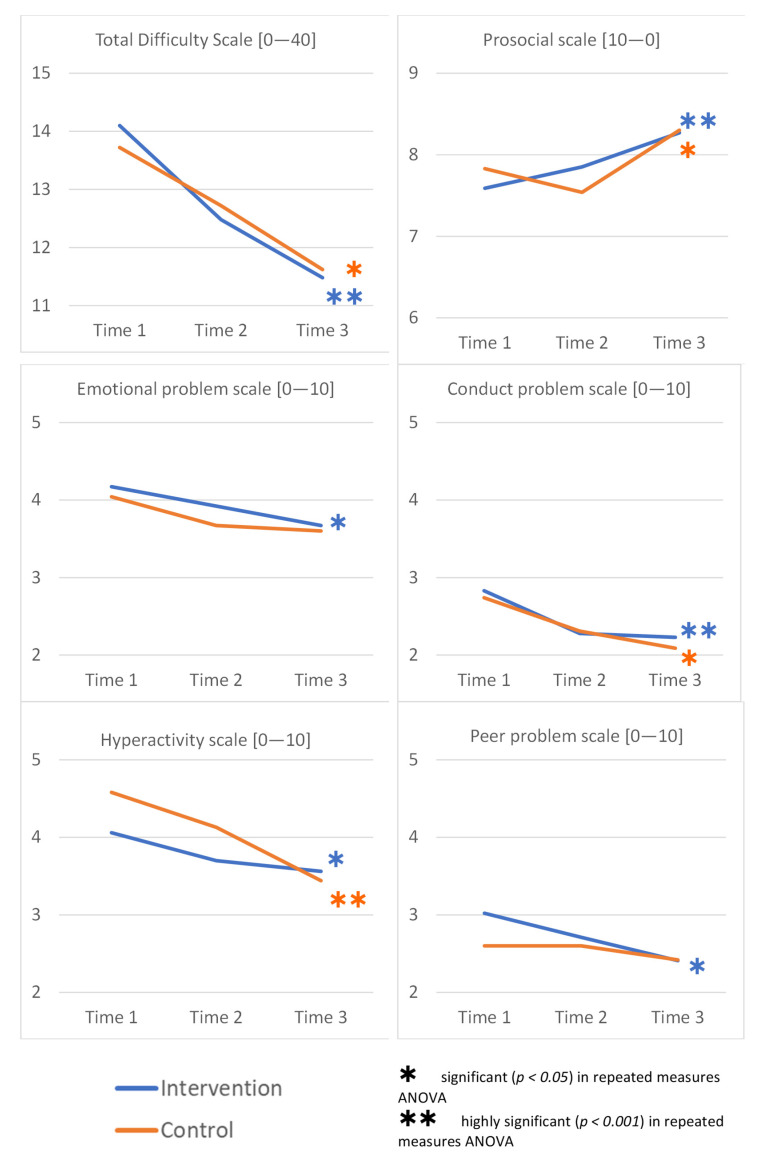
Overall SDQ results comparing intervention and waitlist/control group (*higher scores indicating higher levels of difficulties on all subscales and the Total Difficulty Scale, except for the prosocial scale where higher scores indicate fewer*).

**Table 1 ijerph-18-11137-t001:** Demographic characteristics of study participants within the intervention and waitlist/control group.

Caregiver Demographics		Total (*n* = 289)	Intervention (*n* = 197)	Waitlist/Control (*n* = 92)	*p*-Value	Chi^2^, *t*-Test
		Mean (SD); *n* (%)	Mean (SD); *n* (%)	Mean (SD); *n* (%)		
**Age (in years)**		35.8 (5.86)	35.3 (5.32)	36.7 (6.85)	*0.065*	t_285_ = −1.850
**Gender**	Male	7 (2%)	5 (3%)	2 (2%)	*0.851*	χ^2^ = 0.035
	Female	280 (98%)	192 (97%)	90 (98%)		
**Relationship to child**	Mother	274 (96%)	185 (95%)	89 (97%)	*0.691*	χ^2^ = 2.245
	Father	7 (2.4%)	5 (2.6%)	2 (2.2%)		
	Sister	2 (0.7%)	1 (0.5%)	1 (1.1%)		
	Tutor	2 (0.7%)	2 (1%)	0		
	Other	2 (0.7%)	2 (1%)	0		
**Marital status**	Married	275 (95%)	191 (97%)	84 (91%)	*0.152*	χ^2^ = 6.713
	Divorced	7 (2%)	4 (2%)	3 (3%)		
	Single	1 (0.3%)	0 (0%)	1 (1%)		
	Cohabiting	1 (0.3%)	0 (0%)	1 (1%)		
	Widow	5 (1.7%)	2 (1%)	3 (3%)		
**Education**	Primary school or less	92 (32%)	63 (32%)	29 (32%)	*0.676*	χ^2^ = 3.159
	Some high school	46 (16%)	30 (15%)	16 (18%)		
	Completed high school	79 (28%)	59 (30%)	20 (22%)		
	Trade/technical college qualification	19 (7%)	12 (6%)	7 (8%)		
	University degree	45 (16%)	29 (15%)	16 (18%)		
	Post-graduate	6 (2%)	3 (2%)	3 (3%)		
**Partner’s education**	Primary school or less	105 (38%)	71 (37%)	34 (40%)	*0.619*	χ^2^ = 3.532
	Some high school	51 (18%)	38 (20%)	13 (15%)		
	Completed high school	55 (20%)	38 (20%)	17 (20%)		
	Trade/technical college qualification	16 (6%)	12 (6%)	4 (5%)		
	University degree	38 (14%)	27 (14%)	11 (13%)		
	Post-graduate	14 (5%)	7 (4%)	7 (8%)		
**Work status**	Full time	25 (9%)	19 (10%)	6 (7%)	*0.824*	χ^2^ = 1.517
	Part time	33 (12%)	22 (12%)	11 (12%)		
	Not working but looking for a job	46 (17%)	29 (15%)	17 (19%)		
	Home based paid work	25 (9%)	16 (9%)	9 (10%)		
	Not working	148 (53%)	102 (54%)	46 (52%)		
**Partner’s work status**	Full time	182 (67%)	125 (67%)	57 (67%)	*0.055*	χ^2^ = 7.603
	Part time	53 (20%)	41 (22%)	12 (14%)		
	Not working but looking for a job	22 (8%)	15 (8%)	7 (8%)		
	Not working	15 (6%)	6 (3%)	9 (11%)		
**Country of origin (COO)**	Iran	178 (63%)	122 (65%)	56 (61%)	*0.474*	χ^2^ = 1.493
	Afghanistan	101 (36%)	65 (34%)	36 (39%)		
	Pakistan	2 (1%)	2 (1%)	0 (0%)		
**Years in Iran (if COO not Iran)**		15.1 (13.86)	15.0 (13.47)	15.4 (14.80)	*0.906*	t_102_ = −0.118
**Number of children**		2.2 (1.14)	2.2 (1.17)	2.2 (1.08)	*0.897*	t_277_ = −0.130
Child demographics						
**Age of child taking part in the programme (in years)**		9.7 (1.67)	9.7 (1.62)	9.8 (1.77)	*0.764*	t_286_ = −0.3
**Gender of child in the programme**	Male	126 (44%)	78 (40%)	48 (52%)	*0.048*	χ^2^ = 3.898
	Female	162 (56%)	118 (60%)	44 (48%)		

**Table 2 ijerph-18-11137-t002:** Mean CYRM-R scores over time for families overall and originating from Iran or Afghanistan by intervention or waitlist/control group (*higher scores indicating a higher levels of resilience*).

CYRM-R		Pre-Test Mean (SD)	Post-Test Mean (SD)	Follow-up Mean (SD)	*Two-Way Mixed ANOVA F*(*df_time_*, *df_error_*); *p-Value*; *Partial η^2^*	*Repeated Measures ANOVA F*(*df_time_*, *df_error_*); *p-Value*; *Partial η^2^*	*Post-Hoc Tests*
**Personal resilience subscale** [10–50]							
Iran	Intervention (*n*=53)	41.70 (5.58)	41.55 (5.89)	42.58 (5.10)	F(1.838,130.473) = 0.159; *p = 0.835*; partial η^2^ = 0.002		
	Waitlist/Control (*n* = 20)	42.15 (4.11)	41.55 (4.88)	42.30 (4.32)			
Afghanistan	Intervention (*n* = 53)	40.62 (5.25)	41.02 (5.64)	41.85 (6.81)	F(2,156) = 0.161; *p = 0.851*; partial η^2^ = 0.002		
	Waitlist/Control (*n* = 27)	39.85 (5.95)	40.41 (6.40)	40.63 (7.04)			
**Caregiver resilience subscale** [7–35]							
Iran	Intervention (*n* = 58)	32.26 (2.78)	32.05 (3.70)	32.50 (3.04)	F(2,162) = 0.263; *p = 0.769*; partial η^2^ = 0.003		
	Waitlist/Control (*n* = 25)	32.08 (2.66)	32.36 (2.55)	32.76 (2.26)			
Afghanistan	Intervention (*n* = 52)	29.54 (3.51)	30.23 (3.39)	31.04 (3.40)	F(1.662,129.616) = 1.239; *p = 0.288*; partial η^2^ = 0.016	F(1.655,84.425) = 6.034; *p = 0.006*; partial η^2^ = 0.106	⦿■
	Waitlist/Control (*n* = 28)	29.14 (4.86)	30.11 (3.78)	29.75 (4.53)			
**Overall resilience scale** [17–85]							
Total	Intervention (*n* = 101)	72.17 (7.93)	72.50 (8.29)	73.82 (8.66)	F(1.907,276.449) = 0.155; *p = 0.847*; partial η^2^ = 0.001		
	Waitlist/Control (*n* = 46)	71.22 (8.81)	71.67 (8.61)	72.35 (9.08)			
Iran	Intervention (*n* = 47)	73.87 (7.73)	73.21 (8.62)	75.17 (7.38)	F(2,130) = 0.118; *p = 0.889*; partial η^2^ = 0.002		
	Waitlist/Control (*n* = 20)	74.50 (6.16)	73.65 (6.86)	74.95 (6.00)			
Afghanistan	Intervention (*n* = 48)	70.23 (7.84)	71.42 (8.10)	73.13 (9.36)	F(1.850,133.172) = 0.501; *p = 0.593*; partial η^2^ = 0.007	F(2,94) = 4.070; *p = 0.020*; partial η^2^ = 0.080	■
	Waitlist/Control (*n* = 26)	68.69 (9.77)	70.15 (9.59)	70.35 (10.56)			

Results for repeated measure ANOVAs and post-hoc tests only shown if significant; SD: standard deviation; ⦿ significant difference between t2 and t3, ■ significant difference between t1 and t3.

## Data Availability

The data presented in this study are openly available in the Mendeley Data repository at https://data.mendeley.com/datasets/yfmp8n2n4p/draft?a=2c4cb928-70f4-46d2-8fbd-f742bd384219, accessed on 4 April 2021.

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
