# Peer review of "Impact of a Brief Family Skills Training Programme (“Strong Families”) on Parenting Skills, Child Psychosocial Functioning, and Resilience in Iran: A Multisite Controlled Trial"

_ijerph, 2021, doi:10.3390/ijerph182111137_

Round 1

Reviewer 1 Report

Dear Authors,

I appreciated the chance to read through your manuscript evaluating the Strong Families intervention using a large sample size, with the inclusion of a comparison group to better evaluate changes in the dependent variables (parenting skills, child mental health, and resilience) after the short intervention. While a great deal of data is provided, which is likely to contain findings which could contribute to our understanding of how such interventions can impact parenting and child mental health, as well as child and caregiver resilience, I believe that more extensive and targeted analysis of the data and development of clear interpretations and implications would be necessary to constitute a novel contribution to the field. Please see below for my comments, both major and minor.

Major issues:

  1. Plagiarism: Software has detected that over 20% of the manuscript is copied, verbatim, from other sources, mostly your own (self-plagiarism). This would need to be corrected before a full review could take place, from my perspective. Some of this content is from descriptors in tables, in which case the issue is not as serious. However, attribution and citation is necessary throughout, along with substantial paraphrasing. The necessary parts to paraphrase, quote, or cite are highlighted in an attached PDF.
  2. Ethical considerations: First, the protocol for obtaining parental consent is not clearly detailed in the manuscript, although "caregiver consent" and "child assent" are mentioned in the back matter (IRB statement). However, within the Methods section it is unclear how a potentially vulnerable population was sufficiently informed before giving consent, particularly given the use of a control group. Second, given previous support for the intervention, and the vital importance of the mental health of a vulnerable population, the use of a control group for the collection of data for comparative purposes is not entirely ethical in absence of the provision of the intervention to the "waitlist" participants to receive the intervention following the formal study.

Additional major concerns (by section):

  1. While the issues addressed by your study are important, and constitute a major scale-up from your pilot study (https://doi.org/10.1186/s12889-020-08701-w), there is no clear research gap addressed by these findings, which are generally superficial since they have not been expanded upon to provide clear interpretations and implications. Based on the data analysis methodology and the limited findings, I am afraid that there is no clearly novel contribution within the formulation, implementation, or analysis of the findings, which themselves offer little in terms of implications or clear conclusions, particularly as the intervention is not clearly described.
  1. There is a lack of a single, clear theoretical framework for the development of a clear research gap, with coherent, logical and testable research questions and hypotheses. From the Introduction section, there is no single, clear-cut direction for the study, with the Introduction being largely descriptive in nature, rather than based on an overarching framework to support the evaluation of the proposed intervention with the data collection instruments adopted. For example, the Introduction does not clearly elaborate or operationalize what is meant by mental health or parenting skills, and the construct of resilience is not mentioned once. For research questions to be developed and evaluated, there should be a clear link to the literature (based on a cohesive and consistent theoretical framework), which has not been accomplished.
  2. The organization of the Introduction, Materials and Methods, Results, and Discussion require better use of sub-headings, more sign-posting, and attention to the logic and flow of the writing. Beginning with the introduction, a review of the literature should build towards a specific contribution (the intervention and data collected to evaluate the intervention). This has not been done sufficiently. Organization can flow more logically and be mirrored in subsequent sections.
  1. The Materials and Methods are not adequately elaborated, with some instruments lacking sufficient detail, and the section, overall, lacking in a logical sequence and link to the research questions/Introduction. Weaknesses in this section include: a) a lack of description of the intervention (it is not appropriate to guide readers to reference 26 to find out details regarding the intervention); b) lack of clear details on how informed consent was obtained from a vulnerable population; and c) lack of details on certain data collection instruments, such as the nature of the PAFAS data (in terms of the interpretation of high or low scores, and the range of scores possible).
  2. The data analysis conducted is not rigorous and yields very little in terms of findings, other than some significant improvements in the overall sample which, when analyzed by country, seem to demonstrate far fewer significant differences. While the use of the statistical analysis (two-way mixed ANOVA) is explained well, there is a lack of comparison between countries and no attempts to evaluate possible relationships among dependent variables. For example, the potential influence of parenting-level variables on child-health or resilience variables is not evaluated. This, I believe, could yield meaningful results. However, given the very short intervention, lack of clarity on the nature and process of the intervention, and the lack of effect sizes for most findings, the significance of the results and their generalizability or interpretability are questionable. More rigorous and expanded analysis is recommended. Moreover, interpretation of the results is required, even for what has been found thus far. There is very little text (interpretation) and far too many tables and figures in the Results section. This places the onus on the reader to interpret the findings.
  3. Based on the lack of interpretation of the findings, and a relatively shallow analysis of the data, many of the claims made in the Discussion section are not clearly supported by of the empirical findings, with a great deal of speculation or reliance upon previously published research to interpret quite general points. In fact, the majority of the Discussion section is speculation, rather than clear interpretations, applications, or implications based on the data collected by the study.

Minor issues:

  1. Title: The title is too long and contains a description of the intervention "multisite time-convenience randomized control trial" that is confusing and misleading. The acronym for UNODC should either be written in full or deleted (as the Strong Families programme should suffice). Also, respondents from both Iran and Afghanistan were included in the analysis, but only Iran is mentioned.
  2. Abstract: Greater detail and better structure would be recommended here. Clearer interpretations of the findings are required.
  3. Keywords: "family skills" is an unclear concept and seems to be the same as "parenting." The research methodology might also be used as a keyword.
  4. Introduction: Better structure (using subheadings) is needed. Key constructs (the types of parenting, mental health, and resilience measured by your data collection instruments) should be defined and explained under a consistent framework, preferably based in both theory and empirical findings, such as your pilot study. No research questions or hypotheses appear.
  5. Data collection: subheadings for each instrument are necessary here. There should be sample questions. The interpretation of high or low scores should be clear.
  6. Results: As with most sections, this section should be more concise. Paragraphs (lines 421 to 427) that are not pertinent to the findings should be cut. However, more paragraphs should be added instead of large and unwieldy tables. Table 1 could be briefly summarized and moved to the appendix, if needed. Overall results tend to be significant, while country-specific results tend to show few significant differences. Figure 2, as such, shows significant differences which are not observed in Table 2, for example. There are also no tables for the "overall scores" (such as those depicted in Figure 2), as Table 2 partitions the data by country. Substantial differences between the results of the Iranian participants and Afghani participants (for both intervention and control groups) are obvious, but neither analyzed nor interpreted.
  7. Figures: The figures should be reformatted to be more reader friendly. In their current configuration (without appropriate or consistent minimum/maximum on the y-axis), they are misleading and, in the case of Figure 2, confusing (why are lower scores better?)
  8. Statistics: Effect sizes would be highly informative. In the absence of such data, it is difficult to interpret the findings. Evaluation of relationships among variables, as mentioned above, is strongly encouraged. This would provide a potential contribution. Otherwise, the findings are very limited and offer little beyond your pilot study, at least in the current form.
  9. Table 8: This table could be summarized in a short paragraph, or deleted, since it offers little information to readers.
  10. Discussion: There are no specific findings mentioned, for example by subscale for the main dependent variables. In places, there are claims that are not based in the findings (lines 659 to 674; lines 726 to 730), general findings that are stated without being interpreted (lines 688 to 696), claims should take into consideration other valid interpretations (lines 703 to 710), claims that tend to misrepresent the findings (lines 731 to 740, were it is not mentioned that even for low percentile children, post-test scores were still lower than the overall pre-tests scores for all children), or claims that are too speculative and lacking in any empirical support (lines 746 to 777).
  11. Conclusions: Implications for research, practice, and policy should be mentioned here, in line with the journal's aims and scope.

I wish you all the best in your current and future research!

Reviewer 2 Report

The paper is very interesting and well structured. From a theoretical point of view, the paper is rich in bibliograohical references and clearly and precisaly supports the results obtained. The statistical  analyzes and the study procedure are also well articulated and make the understanding if the report clear. 

However I suggest briefly describing what the "Strong Families" program consists because not all readers may know the features of this intervention program. 

I have seen that a bibliographic reference has been put on it but I think that adding a short description of the program could make your paper more complete. 

Reviewer 3 Report

Thank you for the opportunity to review the manuscript entitled, "Assessing the impact of a brief UNODC family skills training programme (Strong Families) on child mental health, resilience and parenting skills in Iran: A multisite time-convenience randomized controlled trial”.

I believe this study investigated a topic relevant to the readers of “IJERPH”.  The analysis of the mental health of children has become a topic of research of global interest. Overall, it is estimated that between 10% and 20% of children suffer from mental health problems. Parents constitute the main means of development and socialization for most people from a very early age. By the same token, the role of the family unit in psychosocial development is undeniable, with parents being the most powerful force in their children’s lives. I believe in the value of intervention and training programs for families. Preventive intervention programs for parents can help to develop protective mechanisms for children’s development.

The paper addresses important aspects the impact of the Strong Families programme by conducting a trial in Iran, capitalizing on a comparison (wait-list) group. The study involved a multisite non-blinded time-convenience randomized trial with two arms. The results are clearly presented in tables.  The authors obtain some important results and formulate discussions of interest.

In general, this paper is well written and follows well accepted standards of academic writing. However, major revisions may prove beneficial. In this current form the manuscript needs to be improved. 

Introduction:

The introduction not analyze in detail the parentig styles. Differences in parenting styles have been used to account for the effects of familial socialization on children’s social competence; these styles and their differences result from the interaction of different attitudes and behaviors displayed by parents toward their children and they have a direct influence on children’s behavior, emotional security, and well-being.

Rohner (1975) claims that parental acceptance—rejection constitutes a continuum, with parents who express love and affection (both verbally and physically) for their children lying at one end. At the other end, one finds parents who feel aversion for their children, criticize them, and reject them. Affect and communication prevent disorderly conducts among adolescents and stimulate a positive development.

The introduction not analyze in detail the relationship between Parenting and family adjustment skills and Child mental health and a Child resilience. The introduction lacks a clear objectives. There is no relationship between the introduction and some of the results. (“A multiple regression was run to determine how much of the variation of the “total difficulty score” was explained by the number of children in the family, the age of the children and the country of origin”; “Child mental health results in children with high and very high total scores at baseline”, “Child mental health results by gender”; “Child resilience results by country of origin”…)

Instruments:

The authors should explain more clearly the questionnaires (factors, scales and subscales) and the variables objective of the study.

The questionnaires PAFAS and SDQ not present adequate reliability. Alpha reliabilities in the current study are: 0.76 and 0.84 at t1, 0.74 and 0.86 at t2 and 0.74 and 0.86 at t3 (PAFAS).  0.77, 0.77 and 0.80 (SDQ).  Values equal to or greater than .80 are considered acceptable indices of internal consistency.

The reliability of PAFAS, SDQ and CYRM-R was evaluated using Cronbach’s alpha measure. Cronbach's Alpha is conditioned by the number of items and the number of alternative responses, it is necessary to use other alternative reliability indices, such as Composite Reliability (CR) and McDonald's Omega (Ω), which are calculated through factorial loads and are measured more accurate reliability. Furthermore, it is also necessary to estimate the convergent validity using the Extracted Mean Variance (AVE).

Is need the Cronbach’s Alpha, McDonald's Omega (Ω) and Extracted Mean Variance (AVE) to all the factors (scales and subscales): Parental consistency, Coercitive parenting, Positive encouragement, Parent-child relationship, Parental adjustment, Family relationship, Parental teamwork (PAFAS); Emotional problem, Conduct problem, Hyperactivity sacale, Peer problem, Prosocial scale (SDQ); Personal resilience and Caregiver resilience (CYRM-R).

The instruments must always display two important qualities: reliability and validity. It is good practice to perform a confirmatory factor analysis.  Is need to assess the validity of the constructs of the scales PAFAS, SDQ and CYRM-R (Confirmatory Factor Analysis: Relative Chi-Square, P; IFI; GFI; AGFI; CFI; RMSEA).

Statistical Analysis.

The ANOVA analysis (Two-way mixed ANOVA and repeated measures ANOVA) requires key assumptions: independent variables, univariate normality (Test Kolmogorov-Smirnov) and homoscedasticity, the assumption of equality of variances (Test Levene).

A multiple regression (MR) was run to determine how much of the variation of the “total difficulty score” was explained by the number of children in the family, the age of the children and the country of origin.

MR requires key assumptions: linear relationship, multivariate normality, homoscedasticity and finally, the linear regression assumes that there is little or no multicollinearity in the data.

Finally, to eliminate from the dependent variables the effect attributable to variables not included in the design and, therefore, not subjected to experimental control, is need a covariance analysis (ANCOVA), using the control/training group as a fixed factor.

References:

Lack of up-to-date references.

Round 2

Reviewer 1 Report

Dear Authors,

Thank you for responding to the comments and concerns outlined in the previous review. Important improvements have been noted. I will respond to each of the points in the attached file, in a bit more detail.

A few points to consider in addition to those which you have already solved:

a) Reviewing the content for flow, sign-posting, and linking among sections.

b) Stronger integration of your theoretical framework into your description of the Strong Families programme and interpretation of the results in your Discussion section

c) More forceful and clear statement of your contributions, such as in the Conclusions section.

d) More signposting in the Discussions Section

e) Attempting to better format the figures (2 and 3)

Best wishes
